# Tumor Cells Transmit Drug Resistance via Cisplatin-Induced Extracellular Vesicles

**DOI:** 10.3390/ijms241512347

**Published:** 2023-08-02

**Authors:** Jian Wang, Qingqing Liu, Yuanxin Zhao, Jiaying Fu, Jing Su

**Affiliations:** Key Laboratory of Pathobiology, Department of Pathophysiology, Ministry of Education, College of Basic Medical Sciences, Jilin University, 126 Xinmin Street, Changchun 130012, China; wjian21@mails.jlu.edu.cn (J.W.); liuqq22@mails.jlu.edu.cn (Q.L.); yuanxinz22@mails.jlu.edu.cn (Y.Z.); fujy21@mails.jlu.edu.cn (J.F.)

**Keywords:** cisplatin, extracellular vesicles, oxidative stress, non-coding RNA, drug resistance

## Abstract

Cisplatin is a first-line clinical agent used for treating solid tumors. Cisplatin damages the DNA of tumor cells and induces the production of high levels of reactive oxygen species to achieve tumor killing. Tumor cells have evolved several ways to tolerate this damage. Extracellular vesicles (EVs) are an important mode of information transfer in tumor cells. EVs can be substantially activated under cisplatin treatment and mediate different responses of tumor cells under cisplatin treatment depending on their different cargoes. However, the mechanism of action of tumor-cell-derived EVs under cisplatin treatment and their potential cargoes are still unclear. This review considers recent advances in cisplatin-induced release of EVs from tumor cells, with the expectation of providing a new understanding of the mechanisms of cisplatin treatment and drug resistance, as well as strategies for the combined use of cisplatin and other drugs.

## 1. Introduction

Cancer is one of the most prominent health problems worldwide, with approximately 19.3 million new cancer cases and nearly 10 million cancer-related deaths reported in 2020 [1]. Chemotherapy has been used to treat cancer for nearly a century and remains an effective and widely used cancer treatment. Cisplatin is among the most effective metal-based chemotherapeutic agents, from 1965, when Dr. Barnett Rosenberg discovered that cisplatin could inhibit cell division [2], to 1978, when the Food and Drug Administration approved cisplatin for treating testicular cancer, to the present day, when cisplatin is used in nearly 50% of tumor patients [3]. Cisplatin has become the first line of defense against many solid forms of cancer and sarcomas. Studies have confirmed that cisplatin exerts its anti-tumor effects through multiple pathways [4]. Cisplatin usually bind to genomic DNA (gDNA) and mitochondrial DNA (mtDNA), induces DNA cross-linking, prevents DNA replication while blocking mRNA and protein production, and activates multiple transduction pathways that ultimately lead to necrosis or apoptosis [5,6]. The intense DNA toxicity of cisplatin damages the already unstable genome of tumor cells, producing more immunogenic DNA fragments that may play important role in treating tumors with cisplatin [7].

Current evidence suggests that tumor-secreted extracellular vesicles (EVs) are key mediators of intercellular communication between tumor cells in both local and distant microenvironments [8,9]. EVs regulate tumor progression by delivering specific cargoes to induce tumor cell tolerance to multiple therapeutic modalities [10]. Chemotherapy-induced EVs (chemo-EVs) are thought to carry a different cargo from non-chemotherapy-induced EVs. In addition to drug resistance, there is growing evidence that chemo-EVs can determine tumor behavior, particularly metastasis, immune response, and cancer stem cells (CSCs) [11]. Cisplatin-induced immunogenic DNA fragments can be released through the EV pathway [12]. Therefore, it is particularly important to probe the mechanism of the release of EVs from tumor cells induced by cisplatin and the role of the cargo they contain in tumor development, and to adopt a more rational drug combination strategy to improve the therapeutic efficacy of cisplatin.

## 2. Action Mechanism of Cisplatin

Decades of research have elucidated the detailed mechanisms of action of platinum derivatives, such as cisplatin [13,14]. Monohydrated platinum formed after cisplatin enters cells is a strong electrophilic reagent that reacts with nucleophilic reagents, such as sulfhydryl groups of proteins and nitrogenous bases of nucleic acids [15]. This affinity for proteins and bases forms the basis of the mechanism of action of cisplatin. DNA is the primary target of cisplatin’s anticancer activity [16,17]. When cisplatin binds to DNA bases to form crosslinks at a rate exceeding the capacity of the DNA damage repair (DDR) system, cell proliferation is impaired [15,18]. This includes strong inhibition of replicative DNA polymerases to induce apoptosis [18], and cisplatin adducts to promote cell death by inhibiting gene transcription by blocking elongated RNA polymerases [19]. Oxidative stress is another important mechanism by which cisplatin exerts its antitumor effects [20]. Cisplatin reduces intracellular antioxidant levels by strongly binding to sulfhydryl-containing antioxidants and reductases, such as glutathione (GSH) and thioredoxin reductase (TrxR) [21]. Glutathione-platinum (GS-Pt) generated from cisplatin and GSH also inhibit the activity of intracellular thioredoxin (Trx) systems that include Trx, Trx receptor (TrxR), and NADPH [22], which further amplifies damage to the antioxidant system. In addition, the formation of cisplatin-toxic adducts induces the production of high levels of reactive oxygen species (ROS) production, making it easier to overcome the reducing systems present in cells and promoting more severe oxidative stress. High levels of ROS, directly or through cisplatin interactions, cause severe mitochondrial damage and ultimately activate the endogenous mitochondrial apoptotic pathway, accompanied by the activation of autophagy [23,24]. Notably, increased ROS levels due to mitochondrial damage are the primary mechanism underlying cisplatin-mediated nephrotoxicity [25,26,27]. Markus et al. found that cisplatin-sensitive advanced plasmacytoid ovarian cancer cell lines have higher mitochondrial content and higher levels of mitochondrial ROS than cisplatin-resistant cells [28]. These findings suggest that mitochondria play an “effector amplifier” role in cisplatin-mediated oxidative stress and may be a potential way to enhance sensitivity to cisplatin-based anticancer therapies by increasing mitochondrial content or mitochondrial ROS production. In addition, Amélie et al. found that cisplatin inhibits Na+/H+ exchange proteins (NHE) in the membranes of colon cancer cells, leading to intracellular acidosis, increasing membrane fluidity by promoting lipid rafts, and ultimately inducing extrinsic apoptosis via the Fas pathway [29]. Because the membrane is the first barrier that cisplatin must cross to enter cells, NHE may be the first target of cisplatin action, suggesting that cross-linking of cisplatin to proteins may occur earlier than cross-linking with DNA.

## 3. Cisplatin Resistance

The main disadvantage of cisplatin therapy is resistance development in cancer cells. Cisplatin resistance involves four main aspects. First, plasma albumin, transfer protein, and cysteine can strongly bind to cisplatin, leading to the inactivation of a large amount of cisplatin and a significant reduction in the amount of cisplatin that directly enters tumor cells [30,31]. Second, reduced drug accumulation in tumor cells occurs due to decreased cisplatin influx and increased efflux [32,33]. Third, cisplatin is inactivated intracellularly by binding to GSH and metallothionein in cells and is subsequently excreted out of cells [34,35]. Fourth, tumor cells are protected against cisplatin toxicity by overactivation of multiple DDR systems including nucleotide excision repair (NER) and homologous recombination (HR) to promote survival [36,37,38]. Detailed mechanisms of platinum resistance, such as cisplatin resistance, can be found in the excellent review by Zhou et al. [39] and are not repeated here.

## 4. Extracellular Vesicles (EVs)

### 4.1. Overview of EVs

EVs are generally defined as a heterogeneous group of cell-released, 150–1000 nm vesicles with a bilayer structure [40]. EVs include exosomes, microvesicles (MVs), and apoptotic vesicles, which are loaded with proteins, nucleic acids, and lipids, and which are important in intercellular communication [41,42]. The signaling molecules inside EVs are protected by membranes that are less susceptible to enzymatic degradation and can carry messages to more distant sites, making other types of intercellular communication more advantageous. The biogenesis of EVs has been described in detail [43,44,45,46]. Briefly, the biogenesis of exosomes is the inward budding of late endosomes containing intraluminal vesicles (ILVs), which are called multivesicular bodies (MVBs). The cargo of MVBs is partitioned into ILVs. If the product is destined for degradation, the MVBs combine with lysosomes and the cargo is digested. If they act as exosomes, the MVBs will fuse with the plasma membrane and their internal ILVs will be released into the extracellular space [47,48,49]. The inward budding of late endosomal membranes is a critical step in the formation of ILVs, and can depend on either endosomal sorting complex required for transport (ESCRT)-dependent or ESCRT-independent mechanisms required for transport [50]. Four complexes underlie the ESCRT mechanism and are responsible for sorting the cargo to be loaded (ESCRT-0), promoting membrane invagination (ESCRT-I and II), and releasing ILVs in the late endosomal lumen (ESCRT-III) [51]. These findings suggest that proteins and lipids are involved in the ESCRT-independent mechanism of exosome release. For example, Zoraid et al. demonstrated that tetraspanin-rich membrane microstructure domains are involved in exosome biogenesis by aggregating into ordered structures [52]. Inhibition of neutral sphingomyelinase 2 (n-SMase2) has been shown to decrease ceramide levels, thereby reducing exosome release [53], while enzymes that regulate phospholipid and lysosomal phosphatidic acid levels, such as phospholipase, are also involved in influencing the release of exosomes [54].

### 4.2. Chemotherapy Affects the Release of Tumor Cell-Derived EVs

EVs released from tumor cells collectively support tumor progression by participating in tumor microenvironment (TME) formation [55], epithelial–mesenchymal transition (EMT) [56], angiogenesis [57], modulation of the immune system [58], and drug resistance [59]. Notably, tumor cells show enhanced effects on EVs secretion after chemotherapy, which correlates with tumor progression [60,61,62]. For example, Lv et al. found that chemotherapeutic agents such as paclitaxel, irinotecan, and carboplatin apparently increased the levels of HepG2-releasing exosomes [63], and that treatment failure and disease progression in patients undergoing neoadjuvant chemotherapy for breast cancer were associated with elevated levels of EVs [64]. Shyam et al. found that when myeloma cells were exposed to anti-myeloma drugs, they secreted significantly more exosomes and had a different proteomic profile than cells not exposed to the drugs, including acetyl heparinase levels of significantly elevated acetylheparinase, which degrades the surrounding extracellular matrix and translocates between cells, ultimately leading to altered tumor behavior [60]. Helier et al. showed that, compared with normal lung cancer A549 cells, EVs released from constructed cisplatin-resistant lung cancer A549 cells were enriched in extracellular matrix components, cell adhesion protein complement factors, histones, proteasome subunits, and membrane transport proteins [65]. In contrast, patients with acute myeloid leukemia receiving chemotherapy had substantially lower exosomal protein concentrations [66]. Ludwig et al. showed similar results, with lower exosomal protein levels in patients with head and neck cancer treated with oncology [67]. These differences may be due to tumor cell type specificity and drug or time dependence [11]. First, for tumor cells exposed to chemotherapeutic drugs for the first time, it is difficult to resist the intense cytotoxicity, resulting in reduced numbers and lower levels of EVs. Drug-resistant tumor cells formed by prolonged stimulation with chemotherapeutic drugs release large amounts of EVs to remodel the TME in response to survival pressure. For example, Li et al. found that serum levels of miR-106a-5p-containing exosomes were higher in chemotherapy-resistant patients than in non-resistant patients [68]. Second, the conflicting results may also be due to the lack of a harmonized way to quantify EVs from different sources. Finally, the biological properties of the cargo loaded with EVs affect, to some extent, the subsistence of tumor cells, and differences in the number of EVs alone are not sufficient to generalize tumor cells in response to chemotherapeutic agents. Overall, although the chemotherapy-induced release of EVs needs to be studied in more detail, it has been widely proven that tumor cells display a stronger potential for EVs release in response to chemotherapeutic drugs.

### 4.3. Oxidative Stress: A Potential Mechanism by Which Cisplatin Affects the Release of EVs from Tumor Cells

Oxidative stress is a complex cellular process that regulates EVs release. Excess ROS can affect cellular signaling by altering the number and molecular cargo of EVs [51]. Yarana reported that the mechanism by which adriamycin promotes the release of EVs is mainly based on oxidative stress. The concept can be applied to other chemotherapeutic agents, in which oxidative stress is the main mechanism [69]. Similarly, platinum compounds, such as cisplatin, can promote the release of multiple EVs from tumor cells during chemotherapy [70,71], and thus may be related to their induced increase in the level of oxidative stress. Xia et al. showed that cisplatin-induced endoplasmic reticulum (ER) stress induced the release of exosomes encapsulating ER-resident protein 44 (ERp44) from nasopharyngeal carcinoma cells to promote drug resistance in nasopharyngeal carcinoma [72]. Cisplatin-induced increases in total exosomal protein concentration and exosome counts in human ovarian cancer cells (SKOV3) were associated with enhanced levels of oxidative stress and ER stress [73]. In contrast, antioxidants (e.g., thiols or vitamin E) counteract the oxidative stress-induced release of EVs [74,75]. Mechanistically, cisplatin activates p53 [76,77], a major DNA damage response factor that upregulates tumor suppressor activation pathway 6 (TSAP6), an endosomal membrane protein involved in MVB formation [78]. It inhibits the influx of cytoplasmic Ca^2+^ into mitochondria to maintain high levels of cytoplasmic Ca^2+^, which is essential for promoting membrane blebbing and fusion of MVBs with the plasma membrane [79]. Additionally, cisplatin induces autophagy [80,81]. Autophagy is a process closely associated with exosomes biogenesis [82]. Indeed, promoting autophagy will cause MVBs to fuse with lysosomes instead of the plasma membrane, resulting in reduced exosome release [83]. In contrast, inhibiting autophagy increases exosomes release [84,85]. Thus, cisplatin can strongly induce autophagy while promoting exosome secretion, a paradoxical result that deserves further exploration. One of the features of cisplatin-resistant tumor cells is altered vesicular compartment function. This includes a reduction in lysosomal compartments by downregulating Lysosomal-associated membrane protein 1 (LAMP-1) and -2 [86]. Simultaneously, abnormal function and reduced number of lysosomal H^+^ pumps in cisplatin-resistant cells result in reduced lysosomal acidification capacity [87]. This ultimately leads to inadequate lysosomal enzyme processing, affects lysosome number, localization, transport, and fusion, and promotes drug accumulation within lysosomes [86,87,88,89]. Therefore, lysosomal function determines the final outcome of MVB and explains the occurrence of paradoxical phenomena. Structurally and functionally impaired lysosomes in cisplatin-resistant cells may be unable to fully digest the vesicles generated by cisplatin-induced oxidative stress, ultimately leading to the release of high levels of exosomes (Figure 1). This idea is supported by the data obtained by Flora et al. [88]. RAB7 is a small guanosine triphosphatase (GTPase) that plays an important role in several steps of the late endocytic pathway, including endosome maturation, transport from early endosomes to late endosomes and lysosomes, clustering and fusion of late endosomes and lysosomes in the perinuclear region, and lysosomal biogenesis [90]. Down-regulation of RAB7A expression was found to be responsible for the emergence of resistance in cisplatin-resistant cells, mechanistically linked to increased secretion of cisplatin-loaded EVs [88]. In addition, the process of cisplatin-induced oxidative stress that promotes the release of EVs may also be lipid-related. Ceramide is a membrane lipid that deforms the membrane, thereby initiating inward membrane budding [91]. Neutral sphingomyelinase (n-SMase) catalyzes the conversion of sphingolipids to ceramide [92]. Inhibition of n-SMase is usually effective in reducing EV release [53]. Studies have shown that intracellular n-SMase activity is enhanced in the presence of cisplatin, ceramide levels are significantly increased, and total exosome protein concentrations and exosome counts are elevated. These events are associated with increased levels of oxidative and ER stress [73,93]. However, the blockade of n-SMase did not inhibit the release of EVs in any cell type, as demonstrated in the PC-3 prostate cancer cell line [94]. The effect of ceramide may be cell-type-dependent because of the observed variability in the subcellular localization of n-SMase [94,95]. These results suggested that oxidative stress affects the release of EVs, which was closely related to the type of cells and the cell state.

### 4.4. Cisplatin-Induced EVs Released from Tumor Cells May Inhibit Antitumor Effects

As previously described, cisplatin-induced oxidative stress alters the function, location, and aggregation of proteins toxic to the cells. To counteract toxic protein damage, cells initiate protein quality control mechanisms to reduce the accumulation of toxic proteins [96]. The massive release of EVs by cisplatin-induced tumor cells may be a protective mechanism against oxidative damage by scavenging drugs and oxidized proteins. However, these vesicles containing proteotoxic cargoes can be transferred to neighboring or distant cells to trigger intercellular oxidative stress responses. For example, Malik et al. demonstrated that ROS induced after treatment of rat cardiomyocytes with ethanol or transient hypoxia/reoxygenation resulted in the release of HSP60-containing exosomes, which in turn spread to neighboring cells and activated Toll-like receptor 4 (TLR4) in the recipient cells, causing apoptosis in the cardiomyocytes [97,98]. Another issue is that the activation of antioxidant systems or the upregulation of other pro-survival systems must accompany cells under oxidative stress. Thus, these oxidized-protein-containing EVs may act as a “vaccine” for other cells, upregulating the expression of antioxidant systems in advance to protect against drug damage. For example, Eldh et al. reported that hydrogen peroxide (H_2_O_2_) treatment of mast cells leads to changes in the mRNA profiles in the exosomes released from them, and these mRNAs increase the tolerance of mast cells to H_2_O_2_ [99]. Similarly, the release of exosomes containing vascular endothelial growth factor receptor and mRNA for this protein from retinal pigment epithelial cells after ethanol treatment promoted angiogenesis [100]. The reason for this difference in cellular responsiveness to EVs may be related to the type of cells, amount and type of cargo within the EVs, and the duration of action. Considering that tumor cells have a greater ability to cope with injury than normal cells, the oxidative stress products carried by the cisplatin-induced production of EVs may promote the survival or migratory behavior of other tumor cells, or act as damage-associated molecular patterns (DAMPs) to participate in infiltration of immune cells in TME. Priya et al. showed in in vitro experiments that ovarian cancer cells undergo activation of the stress-related c-Jun, N-terminal kinase pathway after cisplatin treatment, while released EVs can induce increased invasiveness and drug resistance in bystander cells, implying that EVs released after cisplatin-mediated stress underlie the induction of further invasion [101]. Similarly, Nelly et al. found that cisplatin modulated the release of exosomes from ovarian cancer cells with CSC characteristics and promoted the tumorigenic activity of bone marrow mesenchymal stem cells (BM-MSCs) [102].

## 5. Cisplatin and the cGAS-STING Signaling Pathway

### 5.1. Cisplatin-Induced Production of Micronuclei (MNi) and mtDNA Is an Activator of Cyclic GMP-AMP Synthase-Stimulator of Interferon Genes (cGAS-STING) Signaling in Tumor Cells

One characteristic of tumor cells is that many chromosomes replicate in the nucleus, which may lead to DNA leakage into the cytoplasm [103]. When the mitotic process is disrupted, some chromatin fragments may be mis-segregated into the cytoplasm, either passively or actively, forming one or more small, spatially separated nuclei called MNi [104]. Thus, MNi is the result of many different endogenous and exogenous damages to DNA and chromosomes. Emary et al. found that most MNi can undergo spontaneous rupture of the nuclear envelope [105] and that ruptured MNi constitutes the major cytoplasmic self-DNA, mainly double-strand DNA (dsDNA) [106]. cGAS is a dsDNA sensor in most mammalian cell types and is responsible for monitoring changes in cytoplasmic DNA content [107]. Notably, it was recently shown that cGAS is constitutively present in the nucleus, and its activation accelerates genomic instability, micronucleus formation, and cell death under stressful conditions by inhibiting the HR pathway [108]. This dual function of cGAS in cytoplasmic lysis, as an innate immune sensor and a negative regulator of DNA repair in the nucleus, emphasizes the importance of cGAS. In addition, numerous studies have shown that mtDNA is a potent inducer of cGAS-STING signaling and can activate the cGAS-STING pathway under a variety of pathogenic conditions [109,110]. This could be because mtDNA has a high copy number, an inefficient repair system, and is easily damaged. Therefore, mtDNA activates cGAS more strongly than nuclear DNA [109]. Mechanistically, binding of dsDNA or mtDNA to cGAS induces dimerization of cGAS and stimulates the enzymatic activity of the cGAS dimer to synthesize 2′3′-cyclic GMP-AMP (cGAMP) from GTP and ATP [111,112]. cGAMP binds to STING as a second messenger and STING subsequently undergoes conformational and positional changes. Recruitment and activation of TANK-binding kinase 1 (TBK1) during STING transfer to the ER–Golgi intermediate compartment, in turn, phosphorylates TBK1 and interferon regulatory factor (IRF3) [113]. IRF3 dimerizes and translocates to activate the type I interferon (IFN) response and interferon-stimulated genes (ISGs).

The activation of the cGAS-STING pathway in tumor cells in response to cisplatin suggests that this signaling pathway may play an important role in tumor progression [114,115]. Cisplatin, a toxic agent targeting DNA, can further exacerbate nuclear and mtDNA damage and cytoplasmic leakage. Vandana et al. found that cisplatin treatment of cutaneous squamous cell carcinoma induced massive MNi formation [116]. Similar results were reported by Otto et al. in breast cancer cells treated with cisplatin [117]. Hiroshi et al. found that cisplatin induced mtDNA leakage into the cytoplasm of tubular epithelial cells and subsequent activation of the cGAS-STING pathway, which triggered inflammation and acute kidney injury in STING-deficient mice and knockout STING renal tubular cells [118]. Similarly, cisplatin dose-dependent escape of mtDNA from the mitochondria was described in cervical cancer cells [119]. Notably, Tigano et al. [120] recently reported that damaged mtDNA acts in synergy with MN to produce a stronger type I IFN response. The authors used the inducible restriction endonuclease AsiSI to selectively induce DNA double-strand breaks (DSBs) in nuclear DNA and radiation to induce DSBs in mtDNA and nuclear DNA. Although AsiSi and radiation caused cells to produce similar levels of cGAS-positive MN (~15%), cells that induced co-damage to nuclear DNA and mtDNA showed stronger ISG activation than cells that induced damage to nuclear DNA alone, with a significant increase in the transcriptional level of ISG. Indeed, compared to bare dsDNA, in vitro reconstructed nucleosomes exhibit higher affinity but lower activation capacity for cGAS [121]. The exact mechanism of how mtDNA interacts with micronuclear DNA to enhance the cGAS-STING pathway remains to be determined, but structural differences between micronuclear DNA and mtDNA (i.e., linear or circular DNA, short or long DNA, with or without histones) appear to be critical. Different regions of cGAS might be involved in sensing micronuclear DNA and mtDNA. For example, the N-terminus of cGAS is required for sensing nuclear chromatin, but not mtDNA [122].

### 5.2. The cGAS-STING Signal: A Double-Edged Sword

Guo et al. reported the increased prevalence of MN in metastatic tumor cells than in primary tumor cells, suggesting that MN may accelerate tumor metastasis [104]. Indeed, nearly 55% of MN derived from metastatic tumor cells were reportedly cGAS-positive [123]. MtDNA from highly metastatic tumor cells to low metastatic tumor cells and stromal cells via EVs may be a novel mechanism for enhanced metastatic potential [124]. Similarly, mtDNA-containing EVs produced by breast cancer cells can enable the intercellular transfer of invasive behavior to promote breast cancer invasion by activating Toll-like receptor 9 (TLR9) in recipient cells [12]. This raises the first question of whether cGAS, as a major recognition factor for MN and mtDNA, is involved in mediating tumor metastasis and/or pro-survival. MN can activate the cGAS-STING pathway and upregulate the downstream noncanonical nuclear factor-kappa B pathway, thereby promoting enhanced fitness and metastasis in chromosomally unstable tumor cells [123]. Vidhya et al. also showed that cisplatin induced mitochondrial Lon, a chaperone and DNA-binding protein that plays a role in PQC system and stress response pathways [125]. Persistently induced ROS can trigger mtDNA damage and its release into the cytoplasm and can induce IFN signaling via cGAS-STING, which in turn can upregulate the expression of programmed death ligand-1 (PD-L1) and immune checkpoint indoleamine 2,3 dioxygenase (IDO-1) to suppress T cell activation. In addition, Lon upregulation reportedly induces secretion of loaded mtDNA and PD-L1 EVs and attenuates innate and CD8+ T cell immunity in the TME by inducing IFN and IL-6 production by macrophages [126]. Grabosch demonstrated cisplatin upregulation in vitro and in vivo via the cGAS-STING pathway PD-L1 expression; the findings support the rationale for the combination of cisplatin with immune checkpoint blockade [114]. Cisplatin can be synergized with PD-1/PD-L1 blocking therapies to improve clinical efficacy; however, a preclinical trial by Liu et al. showed synergistic effects of PD-1 blockade on oxaliplatin-based chemotherapy for gastric cancer, but not on cisplatin-based chemotherapy [127]. This may be because oxaliplatin induces immunogenic death (ICD) in gastric cancer cells more strongly than cisplatin does, making tumor cells more sensitive to immune checkpoint inhibitors targeting PD-1/PD-L1. However, in lung cancer, cisplatin alone reportedly induces the highest levels of ICD-associated DAMPs compared to other chemotherapeutic agents [128]. These clinical findings emphasize the importance of selecting an appropriate ICD-inducing cytotoxin depending on the type of cancer and for the development of chemoimmunotherapy regimens.

Fu et al. described cisplatin-induced dsDNA activation of cGAS-STING signaling inhibiting bladder cancer proliferation by increasing CD8+ T cell and dendritic cell infiltration in a transplanted mouse tumor model [129]. This duality may be related to the degree of cGAS activation under the action of cisplatin, that is, the level of intracellular recognizable DNA fragments. Gusho et al. found substantially high levels of cGAS in cells infected by the dsDNA human papillomavirus that stably maintain viral-free bodies [130]. Moreover, the combination of cGAMP and cisplatin produces a stronger antitumor effect [131]. Therefore, the STING-activated, but not fully activated, state seems to be more dangerous and promotes immune escape from the tumor by reducing the infiltration of cytotoxic T cells (CTL).

The second question is what prevents full activation of the cGAS-STING pathway in the presence of cisplatin. Ubiquitination of cGAS-STING is required to initiate cytoplasmic DNA-mediated activation [132]. Zhang et al. described the high expression of deubiquitinase USP35 in cisplatin-resistant ovarian cancer cells, and demonstrated that USP35 directly deubiquitinated and inactivated STING, but not cGAS, reducing IFN signaling and ultimately leading to reduced infiltration of CD8+ T cells [133]. Similarly, Shoji et al. showed that ubiquinone protein 4 (UBQLN4) delivered STING to the proteasome for degradation during cisplatin treatment and promoted cisplatin resistance in vitro and in vivo [134]. Thus, the targeted degradation of STING by tumor cells “undoes”, at least in part, the efforts of cisplatin-induced cGAS activation, leading to incomplete activation of cGAS-STING in the direction of immunosuppressive TME, which ultimately requires additional STING agonists to achieve the desired therapeutic effect.

However, the consequences of STING degradation are not limited to response impairment of the IFN system. Thomas et al. reported that STING also promoted cell death by regulating ROS and DNA damage, acting as a modulator of cellular ROS homeostasis and tumor cell sensitivity to ROS-dependent DNA damaging agents [135]. Thus, the targeted degradation of STING can not only affect the integrity of the cGAS-STING signaling but also dysregulate the positive feedback system in which cisplatin and STING promote each other, disrupting the tumor-killing effect of cisplatin with STING involvement. Thus, the combination of cisplatin with STING agonists may also be a promising therapeutic approach. Della Corte et al. found that STING activation was associated with higher levels of intrinsic DNA damage, targeted immune checkpoints, and chemokines in patients with primary and recurrent lung adenocarcinoma [136]. Harabuchi et al. increased the gene expression of the chemokines CXCL9 and CXCL10 in tumor tissue using a combination of cisplatin and cGAMP [131], which could promote the recruitment of more CD8+ T cells from the circulation to the TME and achieve the transition from a “cold tumor” to a “hot tumor”.

### 5.3. Activated cGAS and STING Can Be Delivered between Tumor Cells via the EV Pathway

Recent studies have shown that dsDNA and mtDNA can be released outside of tumor cells. Pasquale et al. isolated the entire mitochondrial genome from circulating EVs of cancer patients [137], even intact mitochondria [138]. Dennis et al. also demonstrated the release of dsDNA from tumor cells in an autophagic and MVB-dependent manner [139]. Notably, activated cGAS and STING can also be transferred to recipient cells via the EV pathway (Figure 2). James et al. showed that some dsDNA associated with activated cGAS can be encapsulated within tumor microvesicles (TMVs) in a process regulated by the ARF6 GTP/GDP cycle and described that TMVs can transfer the contents into recipient cells, affecting the behavior of the recipient cells [140]. STING activated by agonists or radiotherapy can undergo RAB22A-mediated atypical autophagy, triggering intercellular transfer to promote antitumor immunity. The RAB22A-induced atypical autophagosome fuses with RAB22A-positive early endosomes, whereas RAB22A inactivates RAB7 and inhibits the fusion of autophagosomes with lysosomes, thus allowing the release of endosomal vesicles with activated STING into the extracellular compartment [141]. Liang et al. showed that when the cGAS-STING signaling pathway is activated, STING oligomers are transported to MVBs and EVs in an ESCRT non-dependent manner, ultimately inhibiting the innate immune response that occurs in cells [142]. Thus, when cGAS-STING is activated, tumor cells may alleviate the damage caused by interferon accumulation by releasing both activated and inactivated STING into the cellular foreign body. The released activated STING may subject the surrounding tumor cells to unwarranted damage and promote antitumor immunity or may be involved in the shaping of immunosuppressive TME, as previously described, with the ultimate outcome depending on which of these STING actions are operative in the tumor cells. However, regardless of the outcome, this direct intercellular transfer of activators may exert a faster and more direct effect than transferring DNA, which is also consistent with the view that the EV pathway is a toxic protein clearance mechanism.

## 6. Non-Coding RNAs (ncRNAs) Are an Important Cargo in EVs Released by Cisplatin-Induced Tumor Cells

The role of ncRNAs in cisplatin resistance in various tumors has been widely demonstrated [143,144]. Three representative ncRNAs are classified according to their length and shape: microRNA (miRNA), long ncRNA (lncRNA), and circular RNA (circRNA) [145]. Recent studies have shown that ncRNAs can be carried in large quantities and released extracellularly in the form of exosomes and are functionally involved in cancer initiation and progression [146]. The types of ncRNAs secreted by tumor cell exosomes in response to cisplatin and their major regulatory roles have been identified (Table 1).

### 6.1. miRNAs Are Key Mediators in Cisplatin Resistance Transmission in Tumor Cells

miRNAs appear to be the most common ncRNAs packaged in exosomes [147]. miRNAs are a group of small endogenous single-stranded ncRNAs, 20–24 nucleotides (nt) in length, that typically inhibit post-transcriptional protein synthesis by binding to the 3′-untranslated region (3′-UTR) of the target mRNA [148]. The action of cisplatin on tumor cells often causes changes in miRNA levels within the tumor cells, and miRNA biosynthetic pathways are critical for maintaining cisplatin-resistant phenotypes [149]. Qin et al. found that cisplatin treatment resulted in the upregulation of miR-182 in human hepatocellular carcinoma (HCC) tissues and HepG2 in HCC cells, and increased resistance to cisplatin [150]. miR-182 can also be enriched and functionally regulated in tumor cell exosomes [151]. A similar phenomenon has been observed in cisplatin-resistant breast and lung cancers [152,153]. These results demonstrate that abnormal alterations in miRNA levels in cisplatin-resistant tumor cells resist the toxic damage caused by cisplatin and also affect neighboring or distant cells in the form of exosomes, which in turn increases the apoptotic threshold of tumor cells, weakens the sensitivity to cisplatin, and promotes migration of tumor cells to distant sites.

### 6.2. Cisplatin and miRNAs, and Oxidative Stress

Less is known about how cisplatin affects changes in miRNA levels in tumor cells. One hypothesis is that this process may be related to cisplatin-induced oxidative stress. Lots of evidence from previous studies shows that miRNA expression is altered by the accumulation of ROS [154]. Transcription factors are upregulated in response to oxidative stress and directly activate miRNA transcription, whereas ROS are directly involved in epigenetic changes, such as DNA methylation and histone modifications, which control specific transcription of specific miRNAs [155]. The ribonuclease Dicer is a key protein in the synthesis of mature miRNAs and is responsible for the cleavage of precursor miRNAs in the cytoplasm [156]. Bu et al. found that small interfering RNA (siRNA) knockdown of Dicer in MCF-7 breast cancer cells resulted in considerable G1 blockade and increased sensitivity to cisplatin [157]. Similarly, Li et al. demonstrated that high expression of Dicer in cisplatin-resistant non-small cell lung cancer (NSCLC) tissues and cells promoted autophagy and drug resistance [12]. In contrast, increased activity of nuclear factor erythroid 2-related factor 2 (Nrf2) in the antioxidant pathway reportedly upregulated Dicer expression [158]. Thus, elevated miRNA levels after cisplatin treatment may be a byproduct of the successful defense of tumor cells against the oxidative toxic damage caused by cisplatin through antioxidant mechanisms, and cisplatin-induced oxidative stress is an important prerequisite for the release of miRNA-containing EVs from tumor cells. Notably, substantially downregulated miRNAs were also detected in cisplatin-treated and cisplatin-resistant cells. miR-363 reportedly directly targeted the 3′-UTR of the anti-apoptotic Bcl-2 family member Mcl-1 [159]. miR-363 expression levels were reduced in cisplatin-resistant HepG2 cells, and downregulation of miR-363 led to Mcl-1 activation in HCC and enhanced resistance to cisplatin [160]. These events may be related to miRNAs and their corresponding transcription factors because the expression of each miRNA gene is controlled by a specific transcription factor [161]. Thus, the status and function of transcription factors under the action of cisplatin may determine the trend of miRNA levels. Admittedly, the regulatory process of miRNA is complex; whether it is early transcriptional events, subsequent shearing and coupling, or epigenetic regulation, cisplatin may affect it. The mechanism of the effect of cisplatin on miRNA needs to be further investigated.

### 6.3. lncRNAs and circRNAs Regulate miRNAs Levels Involved in Cisplatin Resistance

lncRNAs are ncRNAs that are greater than 200 nt in length, and deeply involved in regulating tumor cell resistance to cisplatin [110,162,163]. lncRNAs can act as miRNA sponges by acting as competitive endogenous RNA (ceRNAs) for miRNAs to promote mRNA expression and reduce miRNA regulation of mRNAs [164]. Li et al. found that long-stranded ncRNA uroepithelial carcinoma-associated 1 was upregulated in tissues and serum exosomes of cisplatin-resistant patients and promoted proliferation and resist cisplatin-induced apoptosis through the miR-143/FOSL2 signaling pathway [165]. Similarly, in cervical cancer tissues, lncRNA metastasis-associated lung adenocarcinoma transcript 1 was abundantly expressed in cisplatin-resistant cells and exosomes and inhibited cisplatin-induced apoptosis by targeting miR-370-3p [166]. Given that miRNA and lncRNA interactions are involved in chemoresistance phenotypes, exploring the mechanisms of miRNAs and lncRNAs in chemoresistance may help in the design of cisplatin antitumor therapies to improve patient outcomes. Xie et al. showed that circVMP1 levels were upregulated in cisplatin-resistant NSCLC cell lines compared to cisplatin-sensitive cell lines, and were delivered via exosomes to promote NSCLC progression and cisplatin resistance by targeting the miR-524-5p-METTL3/SOX2 axis [27]. These results suggest that EVs containing lncRNAs and circRNAs secreted by cisplatin-activated tumor cells can be taken up by surrounding tumor cells to propagate drug resistance through regulating miRNAs or proteins in target cells.

## 7. Future Perspectives

Cisplatin is one of the most commonly used anticancer drugs for the treatment of solid cancers (e.g., prostate, ovarian, head and neck, bladder, and lung cancers). However, toxic side effects, drug resistance, and recurrence are the main challenges associated with its clinical use. A series of platinum-based drugs have emerged, but usually have not shown substantial advantages over cisplatin. Deeper understanding of the mechanism of action and resistance to cisplatin may facilitate the development of more effective new drugs or provide new therapeutic strategies that combine cisplatin. The use of cisplatin in combination with other drugs has shown some promise. The Food and Drug Administration and European Medicines Agency have also approved PD-1 blockade drugs in combination with platinum-based chemotherapy. Targeting EVs released during cisplatin treatment may also be a potential strategy, as cisplatin induces tumor cells to secrete a large number of EVs with different expression profiles to propagate resistance. For example, curcumin can partially modulate the composition of EVs released from cisplatin-resistant tumor cells to inhibit the development of cisplatin resistance [167]. However, similar to the combination of cisplatin and STING agonists, these regimens are currently in the experimental stage of research. More effective and safer drug combinations need to be further explored in large clinical trials. In addition, since ncRNAs are key regulators and predictors of cancer therapeutic resistance and can be released via the EV pathway, they can be used as therapeutic adjuvants and components of genetically and molecularly characterized therapeutic strategies targeting tumors to improve the anticancer response to existing therapeutic modalities. Despite the remarkable progress in the field of ncRNA-based therapeutics, there are still many challenges to be addressed. These include side effects caused by off-target effects, as well as unexpected target effects when administered systemically to normal tissue, rather than tumor tissue. Therefore, the specificity, delivery, and tolerability of therapeutic approaches using ncRNAs require further improvement. Concerning the heterogeneity of ncRNAs in tumors, the development of single-cell spatial non-coding transcriptomics and single-cell sequencing technologies for ncRNAs will help address this issue. In a different tack, the use of nanoparticles as a delivery system can increase the homogeneous distribution and accumulation of drugs within the target tumor tissue. This approach offers considerable advantages in reducing off-target effects and normal tissue damage. Nanoparticles can also be directly applied to the transport of platinum-based drugs, such as cisplatin, to limit their toxicity by increasing targeting to tumor cells and reducing their accumulation in normal cells. Craig et al. doped cisplatin into gold nanoparticles via a thioglycol linker, which improved targeting and cellular uptake, with more than 110-fold higher cytotoxicity than cisplatin against A2780 and A2780/cp70 cancer cell lines [168].

Targeted drug combination therapy, based on the knowledge that cisplatin treatment affects the release of EVs from tumor cells, will help expand the clinical application of cisplatin. The refinement of more advanced drug delivery systems will allow cisplatin, a classic anticancer drug, to continue to shine in the treatment of cancer.

## Figures and Tables

**Figure 1 ijms-24-12347-f001:**
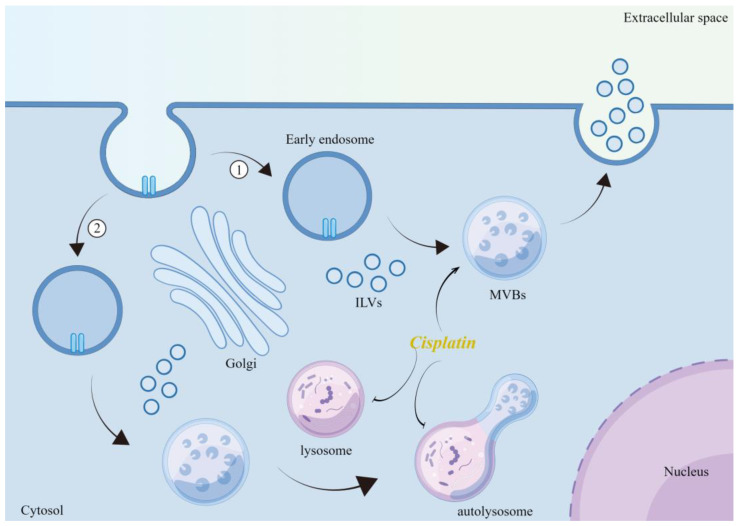
Mechanism of cisplatin-induced release of EVs. Cisplatin promotes the formation of MVBs through oxidative stress, while inhibiting the maturation of lysosomes and the formation of autophagic lysosomes, leading to the release of increased ILVs into the extracellular space. Illustration was made using Figdraw.

**Figure 2 ijms-24-12347-f002:**
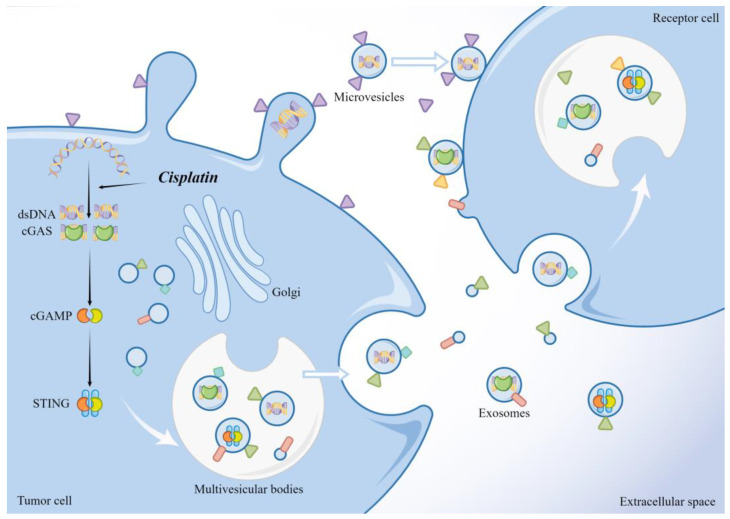
dsDNA, activated cGAS, and STING in tumor cells can be released through the EV pathway and taken up by the recipient cells to exert their effects. The illustration was made using Figdraw.

**Table 1 ijms-24-12347-t001:** Tumor cells secrete ncRNAs as exosomes in response to cisplatin.

ncRNA	Expression	Target	Function	PMID
Circ_0074269	+	miR-485-5p	Anti-apoptosis	35075616
Circ_0063526	+	miR-449a	Promote migration, invasion and autophagy	36206102
Circ Foxp1	+	CEB pg and FMNL3	Promote proliferation and cisplatin (DDP) resistance	32808501
Circ PVT1	+	miR-30a-5p	Anti-apoptosis, Promote migration and autophagy	32799541
Circ VMP1	+	miR-524-5p	Promote proliferation, migration, invasion, and DDP resistance	35467477
Circ cdr1a	-	miR-1270	Anti-apoptotic	31479922
lncRNA HOTTIP	+	miR-218	DDP resistance	31908497
lncRNA PICSAR	+	miR-485-5p	DDP resistance	33817237
lncRNA HEIH	+	miR-3619-5p	Promote proliferation and DDP resistance	33130420
lncRNA UCA1	+	miR-143	Anti-apoptosis	31234009
miR-193b-3p	+	ZBTB7A	Promote proliferation and Anti-apoptosis	36155593
miR-769-5p	+	CASP9	Anti-apoptosis	35522909
miR-425-3p	+	AKT1	Activate autophagy process	31632022
miR-4443	+	METTL3	Anti-apoptosis and Ferroptosis	33781830
miR-155	+	FOXO3a	Promote EMT	32284792
miR-643	+	APOL6	Anti-apoptosis	34071504
miR-27a	+	P53	Anti-apoptosis	31153637
miR-21	+	PDCD4	Promote invasion	30840273
miR-1273a	-	SDCBP	Anti-apoptosis	32901857
miR-30a	-	Beclin1	Anti-apoptosis	32499869

## Data Availability

Not applicable.

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
