# Peer review of "Tumor Cells Transmit Drug Resistance via Cisplatin-Induced Extracellular Vesicles"

_ijms, 2023, doi:10.3390/ijms241512347_

Round 1

Reviewer 1 Report

Here, the authors present a comprehensive review summarizing the roles of exosomes in cisplatin resistance. After a very broad introduction, the authors comprehensively reviewed mechanisms of cisplatin-based therapy, detailed knowledge of extracellular vesicles (exosomes), and the role of cGAS-STING pathway in cisplatin-based therapy. Knowldege of non-coding RNAs is summarized in text and a table. They end with a (quite lenghty) conclusion and perspective section.

In general, the authors work is highly appreciated giving a very detailed overview of a topic which is very relevant to the field. However, I feel that the manuscript - even for a review article-  is very broad and lenghty (almost 20 pages and over 200 references!). I highly recommend a more specific focus of the article, also regarding a number of already existing very good review articles providing information about cisplatin-based therapy and exosomes in general (authors can refer to that instead of writing it again). In summary, I would recommend this review article for publication in IJMS with minor revisions.

I have listed some points that the authors should address before publication. In my opinion this will increase readability and interest to a broad readership!

- editing of english is required, many formal errors, punctuation, use of upper case and so on.

- all sections should be shortened and focusing on most important findings

- number of references is also very high, and should be revised!

- section conclusions and perspectives should be much shorter concentrating on 1-2 ideas for future directions (here again some previous findings are discussed which should be placed in the text before)

english should be revised as stated above (many formal errors, punctuation, use of upper case and so on)

Author Response

请看附件

Reviewer 2 Report

This review can be interesting to scientists working in the field of medicinal inorganic chemistry. It gives an overview of the possible mechanisms of resistance to cisplatin by cancer cells. The review is well organized in the part that describes extracellular vesicles and responses to cisplatin treatment, however it is necessary to shorten the introduction about the cisplatin mechanism of action (ligand exchange, binding to albumin, and binding to DNA), as it is well known, has been reviewed many times, and does not need to be repeated again. I would recommend accepting this review after the reorganization of the text.

Author Response

请看附件

Round 2

Reviewer 2 Report

The revised version is now sutable to be aceepted.